



# Barotropic vorticity balance of the North Atlantic subpolar gyre in an eddy-resolving model

Mathieu Le Corre[1], Jonathan Gula[1], and Anne-Marie Tréguier[1]

[1]Univ. Brest, CNRS, IRD, Ifremer, Laboratoire d'Océanographie Physique et Spatiale (LOPS), IUEM, Brest, France

**Correspondence:** Mathieu Le Corre (mathieu.lecorre@univ-brest.fr)

**Abstract.** The circulation in the North Atlantic Subpolar gyre is complex and strongly influenced by the topography. The gyre dynamics is traditionally understood as the result of a topographic Sverdrup balance, which corresponds to a first order balance between the planetary vorticity advection, the bottom pressure torque and the wind stress curl. However, this dynamics has been studied mostly with non-eddy-resolving models and a crude representation of the bottom topography. Here we revisit the

barotropic vorticity balance of the North Atlantic Subpolar gyre using a high resolution simulation ($\approx$ 2-km) with topography-following vertical coordinates to better represent the mesoscale turbulence and flow-topography interactions. Our findings highlight that, locally, there is a first order balance between the bottom pressure torque and the nonlinear terms, albeit with a high degree of cancellation between each other. However, balances integrated over different regions of the gyre – shelf, slope and interior – still highlight the important role played by nonlinearities and the bottom drag curls. In particular the topographic

Sverdrup balance cannot describe the dynamics in the interior of the gyre. The main sources of cyclonic vorticity are the non linear terms due to eddies generated along eastern boundary currents and the time-mean nonlinear terms from the Northwest Corner. Our results suggest that a good representation of the mesoscale activity along with a good positioning of the Northwest corner are two important conditions for a better representation of the circulation in the North Atlantic Subpolar Gyre.

## 1 Introduction

The North Atlantic Subpolar Gyre (SPG) is a key region for the meridional overturning circulation (MOC). There, the North Atlantic surface waters coming from the subtropical gyre are transformed into denser waters that flow southward and form the lower limb of the MOC. The dynamics of the currents in the SPG is a result of strong buoyancy gradients, intense surface buoyancy and wind forcings, and exchanges of waters with the Nordic Seas through overflows. Understanding this complex dynamics is essential to better understand the mechanisms that drives the variability of the MOC.

The dynamics of wind-driven oceanic gyres is traditionally understood as the result of two distinct balances for the interior of the gyre and the boundary of the gyre, where currents flow along topography. In the interior, the flow follows a Sverdrup balance, which corresponds to a first order balance between the wind stress curl and a meridional transport in the barotropic (depth-integrated) vorticity balance. This balance has been shown to hold in the interior of subtropical gyres (Hughes and De Cuevas, 2001; Thomas et al., 2014; Yeager, 2015; Schoonover et al., 2016; Sonnewald et al., 2019; Le Bras et al., 2019).

Where the currents interact with the topography, another term becomes first order in the barotropic vorticity balance: the



Bottom Pressure Torque (BPT). The BPT includes the impacts of the bottom topography on the barotropic currents, and derives from the interaction of the abyssal geostrophic flow with the sloping bottom bathymetry. Works by Hughes (2000); Hughes and De Cuevas (2001); Jackson et al. (2006); Schoonover et al. (2016) have demonstrated the prevalence of the BPT in the global barotropic vorticity balance. They have shown in particular that the BPT is the dominant term in western boundary

currents, thus demonstrating that viscous effects were not required to close the vorticity budget of the gyres as hypothesized in the classical works of Munk (1950). The SPG circulation is strongly shaped by the bottom topography. Due to the low stratification, the currents have a strong barotropic component (Van Aken, 1995; Daniault et al., 2016; Fischer et al., 2004). They are thus strongly impacted by the steep topography around the gyre. The importance of the bottom topography in driving the SPG dynamics emerged quite early in the works of Luyten et al. (1985) and Wunsch (1985). The prevalence of the BPT in

the SPG has also been demonstrated by Hughes and De Cuevas (2001); Spence et al. (2012); Yeager (2015). All studies also pointed out a failure of the flat bottom Sverdrup balance in this area.

The studies putting forward the importance of the BPT in the SPG have been using coarse resolution models. But currents in the SPG are also strongly influenced by eddies, which can modify the mean flow structure (McWilliams, 2008). Models then require resolutions able to resolve these effects. Eddy-permitting resolutions have been shown to improve the characteristics

of the boundary currents of the SPG, including a better position of the currents, narrower lateral extensions and velocity amplitudes closer to observations (Treguier et al., 2005; Danek, 2019). The vertical structure of the currents is also improved with a more barotropic structure for the boundary currents around the SPG (Marzocchi, 2015). These changes, compared to coarser resolution models, allow the inertial effects to become more important and modify the interactions with the topography. Also, at higher resolution, the viscosity is reduced and the bottom topography and inertial effects become prevalent, allowing

the flow to better match the observations (Spence et al., 2012; Schoonover et al., 2016).

Recently, Sonnewald et al. (2019) clustered regions dominated by different barotropic vorticity balances using a global $1° \times 1°$ model. They retrieved the results of a SPG dominated by BPT effects, but also a part of the gyre dominated by Non-Linear (NL) effects, despite the relatively coarse resolution of the model. Yeager (2015) compared results from a $1°$ resolution model with an eddy-permitting $1/10°$ resolution model, and noticed an increase of the amplitude of the NL term by a factor 3

in some locations. However, it did not modify significantly the first order equilibrium between the wind, planetary vorticity and BPT. The impact of the NL term becomes clearer at higher resolution. With a $1/20°$ resolution simulation Wang et al. (2017) showed the importance of this term in the dynamics of recirculation gyres such as the Gulf Stream recirculation gyres or the North Western Corner.

In addition to the horizontal resolution, the representation of the bottom topography has an impact on the structure of the

flow. z-level coordinates have the tendency to create too shallow flows compared to partial step coordinates (Pacanowski and Gnanadesikan, 1998). Terrain following coordinates ($\sigma$-level) have proven effective in representing boundary currents (Schoonover et al., 2016; Ezer, 2016). The z-level types coordinates tend to have too much viscosity and/or diffusivity close to the topography due to the presence of vertical walls. This effect is corrected when increasing the vertical resolution or using partial steps to converge to results obtained with $\sigma$-coordinates (Ezer and Mellor, 2004).



The aim of this paper is to investigate the dynamics of the SPG by analysing the barotropic vorticity balance in a truly eddy-resolving $\sigma$-level coordinate model. To our knowledge no study of the SPG dynamics has ever been conducted at this resolution with this kind of vertical coordinates. The switch in vertical coordinates combined with eddy-resolving resolution might help to resolve smaller scale processes and allow a better representation of flow-topography interactions overall. The paper is organised as follows: The simulation setup is presented in section 2. The mean currents characteristics and variability

in the simulation are confronted to observations in section 3. The barotropic vorticity balance is analyzed for the full SPG in section 4. The balances corresponding to the different parts of the gyre are further described in section 5. To better understand what is hidden inside the non linear term we analuze it more in details in section 6. Conclusions are presented and discussed in section 7.

## 2    Model and set-up

To investigate the impact of the topography on the circulation, it is essential to have a good representation of the flow-topography interactions. To do so, we use a terrain-following coordinate model: the Regional Oceanic Modelling System (ROMS, Shchepetkin and McWilliams (2009)) in its CROCO (Coastal and Regional Ocean Community) version (Debreu et al., 2012). It solves the hydrostatic primitive equations for velocity, temperature and salinity, using a full equation of state for seawater (Shchepetkin and McWilliams, 2009, 2011).

To achieve a kilometric resolution at a reasonable cost, we use a one way nesting approach by defining two successive horizontal grids with resolutions $\Delta x \approx 6$ km for the parent grid covering the North Atlantic ocean (NATL) and $\Delta x \approx 2$ km for the child grid covering the SPG (POLGYR). The parent North Atlantic domain is identical to the one in Renault et al. (2016). It has $1152 \times 1059$ points with a horizontal resolution of 6—7 km. The child grid has $2000 \times 1600$ points and a horizontal resolution of 2 km. It allows the simulation to be truly eddy resolving in most of the area, as the Rossby deformation radius

varies between 10 and 20 km over the region (Chelton et al., 1998). The domains are shown in figure 1.

    The bathymetry for both domains is constructed from the SRTM30 PLUS dataset (available online at http://topex.ucsd.edu/WWW_html/srtm30_plus.html) based on the 1 min Sandwell and Smith (1997) global dataset and higher resolution data where available. A Gaussian smoothing kernel with a width 4 times the topographic grid spacing is used to avoid aliasing whenever the topographic data are available at higher resolution than the computational grid and to ensure the smoothness of

the topography at the grid scale. Also, to avoid pressure gradient errors induced by terrain-following coordinates in shallow regions with steep bathymetric slopes (Beckmann and Haidvogel, 1993), we locally smooth the bottom topography $h$ to ensure that the steepness of the topography does not exceed a factor $r = 0.2$, where the local $r$-factor is defined in the x and y directions by $r_x = \frac{h(i,j) - h(i-1,j)}{h(i,j) + h(i-1,j)}$ and $r_y = \frac{h(i,j) - h(i,j-1)}{h(i,j) + h(i,j-1)}$, (i,j) representing the grid index.

    Initial and lateral boundary data for the largest domain are taken from the Simple Ocean Data Assimilation (SODA, Carton

and Giese (2008)). The NATL simulation is run from January 1st, 1999 to December 31st, 2009. It is spun up for 2 years, and the following 8 years are used to generate boundary conditions for the child grid. Our focus is the barotropic vorticity dynamics, characterized by time scales on the order of months, such that a year of spin up is sufficient for the kinetic energy to





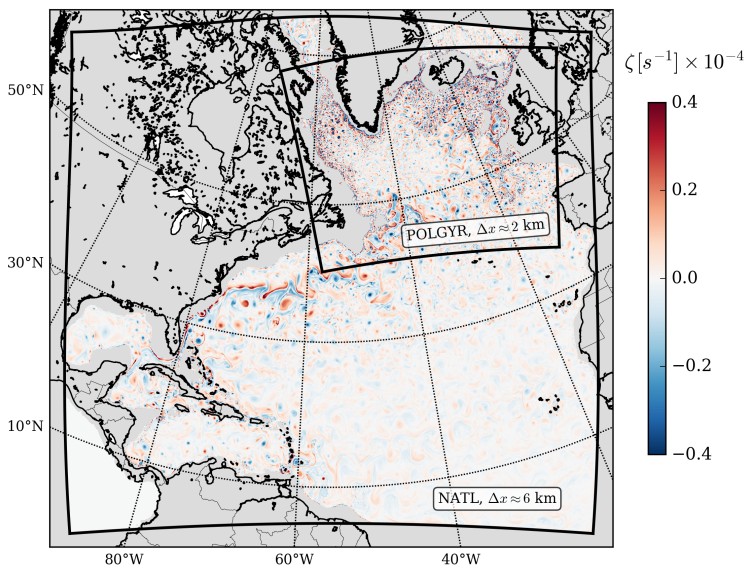

**Figure 1.** Snapshot of the relative vorticity at 500 m depth in the North Atlantic in the NATL simulation. The NATL grid ($\Delta x \approx 6$ km) covers most the North Atlantic, and the POLGYR grid (smaller rectangle, $\Delta x \approx 2$ km) covers the subpolar gyre.

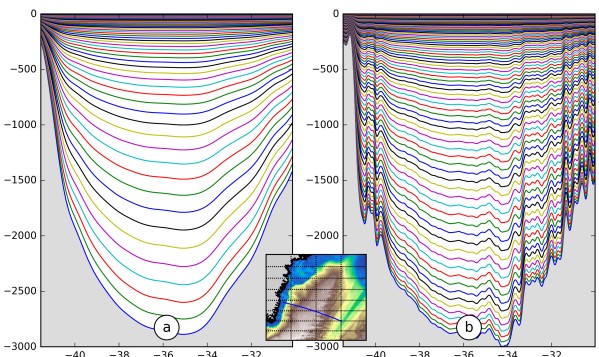

**Figure 2.** Depths of the model vertical $\sigma$-levels along a section of the Irminger Basin for (a) the 6-km simulation (NATL), and (b) the 2-km simulation (POLGYR).

reach a state of quasi-equilibrium in POLGYR (not shown). The study is carried on the 7 remaining years between 2002 and 2009. The surface forcings are daily ERA-INTERIM data for the parent grid and 12-hourly ERA-INTERIM data for the child grid.

The North Atlantic and subpolar gyre simulations have 50 and 80 vertical levels, respectively. Vertical levels are stretched at the surface and bottom (Lemarié et al., 2012) to have a better representation of the surface layer dynamics at the top and flow-topography interactions at the bottom. The depth of the transition between flat z levels and terrain-following $\sigma$ levels is





$h_{cline} = 300$ m. The two parameters controlling the bottom and surface refinement of the grid are $\sigma_b = 2$, $\sigma_s = 7$ for the parent
grid and $\sigma_b = 3$, $\sigma_s = 6$ for the child grid, corresponding to strongly stretched levels at the surface and bottom (Figure 2).

The vertical mixing of tracers and momentum is done by a k-$\epsilon$ model (GLS, Umlauf and Burchard (2003)). The effect of
bottom friction is parameterized through a logarithmic law of the wall with a roughness length $Z_0 = 0.01$ m.

## 3 Mean Currents and variability

### 3.0.1 Mean circulation

Before investigating what is driving the SPG dynamics, we first need to validate the mean circulation in our simulations. Mean
velocities from the two simulations (NATL and POLGYR) at the surface and 1000-m depth are shown in figure 3. We present
at the bottom of figure 3 (e,f) the amplitudes of the currents from the NOAA drifter climatology (Laurindo et al., 2017) at the
surface and from the ARGO-based ANDRO dataset at 1000-m depth (Ollitrault and Rannou, 2013; Lebedev et al., 2007). The
ANDRO data have been binned on a $0.25° \times 0.25°$ grid and cells with less than 10 data points have been removed.

The North Atlantic Current (NAC) represents a boundary between the subtropical and the subpolar gyres. Oceanic models
have difficulties in reproducing its dynamics and particularly its Northern extension known as the NorthWest Corner (Bryan
et al., 2007; Hecht and Smith, 2008; Drews et al., 2015), which is centered at 50° N, 48 ° W (Lazier, 1994). These difficulties
lead to the apparition of the so called "cold-bias", which can reach up to 10 °C (Griffies et al., 2009; Drews et al., 2015),
and which plays a role in the Atlantic low frequency variability (Drews and Greatbatch, 2017). The NorthWest Corner is well
reproduced in our simulations, and the temperature bias at this location is less than a degree.

After turning eastward, the NAC splits into three branches, which are strongly constrained by topography (Bower, 2008).
They cross the Mid Atlantic Ridge (MAR) through three deep fracture zones: the Charlie-Gibbs Fracture Zone (CGFZ, 52.5 °
N), the Faraday fracture zone (50° N) and the Maxwell fracture zone (48° N) (Bower et al., 2002). In both surface and 1000
m observations (Fig. 3(e),(f)), the Northern branch of the NAC is more intense and corresponds to the main pathway across
the MAR. The three branches are well represented in the simulations with, at the surface, an overestimation of the southern
branch and an underestimation of the northern branch. At depth, ANDRO data depict an intense branch crossing the MAR
at the CGFZ while the amplitude of the two Southern branches is smaller. This feature might be related to the Labrador Sea
Water passing into the Eastern Basin through the CGFZ in this depth range, while in the Faraday and Maxwell Fracture zones
the flow is more surface intensified. The circulation in POLGYR is closer to the observations with a better representation of
the flow in the CGFZ at 1000 m.

After crossing the MAR, the three branches head North with the two Northern ones feeding the interior of the Iceland basin
and the Rockall Trough (RT) (Daniault et al., 2016). The water coming from the Maxwell fracture zone recirculates southward
in the West European Basin (Paillet and Mercier, 1997). As most of the models (Treguier et al., 2005; Deshayes et al., 2007),
NATL and POLGYR are consistent with observations for the circulation in the Eastern Basin with a good positioning of the
two main branches passing respectively in the Maury channel (deepest part of the Iceland Basin west of Hatton Bank) and the
RT.





**Figure 3.** Mean velocity averaged over 2002-2008 at the surface (left) and 1000-m (right) in NATL (a,b), POLGYR (c,d) and observations, NOAA drifters and ANDRO (e,f).





A deep permanent anticyclonic eddy is found in Rockall Trough (**?**Smilenova et al., to be submitted; Le Corre et al., 2019). This structure is detectable in the ANDRO dataset around 55° N, 12° W (Fig. 3(f)). It is not present in NATL while it appears in POLGYR, albeit with too intense velocities. In NATL at depth, there is a strong southward flow in the western part of the RT

due to the wrong representation of the Faroe Bank channel. As the topography is strongly smoothed, the channel is not properly represented and does not allow the dense water coming from the Nordic Seas to pass through it and feed the Iceland Scotland Overflow Water properly (Hansen et al., 2016; Kanzow and Zenk, 2014). Thus, the water is recirculating in the western part of the RT, creating a spurious pattern (Fig. 3(b)). The problem is solved by increasing the horizontal resolution and improving the representation of the topography, which corresponds to a wider opening of the channel and allows a more realistic circulation

in the RT.

Further north, part of the flow continues to the Nordic Seas (Rossby and Flagg, 2012), while the other part follows the Reykjanes Ridge (RR). A common bias in models east of RR is a too intense southward flow at the surface (Treguier et al., 2005). This bias is present in NATL but disappears at higher resolution in POLGYR, which is closer to the circulation observed by the drifters. On the western side of the RR the signal of the strong northward Irminger current visible in observations is well

resolved by the simulations (Fig. 3).

At 1000-m depth, Argo floats reveal a continuous current following the Eastern RR flank until reaching the CGFZ, with some of the flow crossing the ridge North of 57.3° N and some crossing at the Bight Fracture Zone (56-57° N). This is coherent with the results from Petit et al. (2018), which observed that water at this depth (their layer 3) was more likely to cross the ridge North of 56°N. This southwestward flow is present in our simulations, with too intense velocity amplitudes in NATL, but

realistic amplitudes at higher resolution in POLGYR. In both cases, we clearly see the flow crossing the ridge north of 56°N. On the western side of RR, the velocity in the simulations is too strong compared to observations. The mean subpolar gyre intensity in the model (Fig. 4), computed as the cumulative transport from Iceland to 53.15° N along the crest of the RR, is equal to -25 Sv and compares well with the -21.9 ± 2.5 Sv monthly average in Petit et al. (2018).

Numerous recirculations are present in the SPG, many of them occurring near the intense boundary currents along Greenland

and around the Labrador sea (Reverdin, 2003; Flatau et al., 2003; Cuny et al., 2002). The recirculation cells are present in the Labrador sea (Lavender et al., 2000; Cuny et al., 2002) and extend to the Irminger basin (Holliday et al., 2009). Theses features are mainly driven by the topography and the wind as described in Käse et al. (2001); Spall and Pickart (2003), and are stable in time (Palter et al., 2016). Some models are unable to reproduce correctly the recirculation cells, especially the one in the center Labrador Sea (Treguier et al., 2005). In our case, this recirculation is well represented (Figure 3(a),(b),(c),(d)). The

counter current flows offshore the Labrador continental slope, with a North extension at 60° N, which matches observations from Lavender et al. (2005). At the tip of Greenland, this counter current separates in two to form a branch flowing inside the Irminger Basin while the other branch is redirected southward. This second branch is relatively intense in our simulation but is also present in ANDRO data (Fischer et al. (2018), their Figures 3 and 5a).



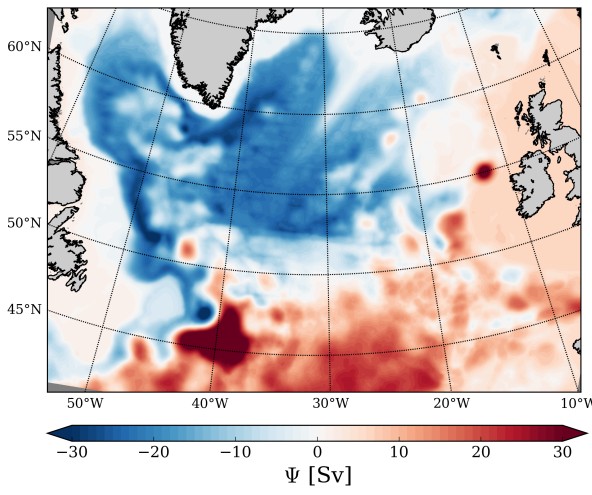

**Figure 4.** Time mean barotropic stream function over 2002-2008

## 3.1 The mesoscale activity

The mesoscale activity plays a big role in redistributing water masses properties in the SPG (de Jong et al., 2016; Zhao et al., 2018). The presence of mesoscale eddies can be inferred by their signatures on the Eddy Kinetic Energy (EKE). From the surface EKE signal extracted from NOAA drifters on a $0.25° \times 0.25°$ grid (Fig. 5(e)), we retrieve the main hot spots described by Flatau et al. (2003) in the SPG: the Labrador Sea, the Irminger and Iceland basins. Those signals are mainly due to generation of mesoscale eddies through baroclinic and barotropic instabilities of the boundary currents.

EKE amplitudes in the NATL simulation are weaker than in observations, but the eddy activity is enhanced when the resolution is increased. The POLGYR simulation displays similar EKE patterns than observational data in every basins (Labrador, Irminger and Iceland) with close amplitudes over most of the SPG. The EKE patterns corresponding to the generation of Irminger Rings have higher magnitudes in POLGYR than in the NOAA drifters data.

A way to quantify the mesoscale activity at depth is to look at the vertical isopycnal displacements. When referenced to a 175 mean, it represents the Eddy Available Potential Energy (EAPE) or the amount of energy stored in the potential energy reservoir due to mesoscale activity (Lorenz, 1955). This quantity is a proxy of the baroclinic activity in the interior of the ocean. We compare EAPE from the simulations with the atlas of Roullet et al. (2014) constructed from Argo data (Fig. 5(f)). In NATL (at 6 km resolution) most of the baroclinic activity already seems well resolved. However, observations highlight an EAPE maximum on the western flank of the RR that is missing in NATL, but appears only in POLGYR (at 2 km resolution). On 180 the contrary, strong patches of EAPE are visible along the boundary currents of the western half of the SPG in NATL, but are not visible in observations. Interestingly these patterns weaken in POLGYR, potentially pointing to a change in the vertical structure of the currents at higher resolution. Another factor to take into consideration is the lack of Argo measurements close to the boundaries, which might cause an underestimation of EAPE at these locations.





**Figure 5.** Mean Surface Eddy Kinetic Energy (left) and Mean Eddy Available Potential Energy between 2002 and 2008 (right) in NATL (a,b), POLGYR (c,d). There are compared with result from the NOAA database (e) and the EAPE Atlas from Roullet&al (f)



## 4 Vorticity balance of the subpolar gyre at high resolution

### 4.1 An overall view of the subpolar gyre vorticity balance

The barotropic vorticity equation is obtained by integrating the momentum equations in the vertical and cross-differentiating them (Gula et al., 2015):

$$
\underbrace{\frac{\partial \Omega}{\partial t}}_{rate} = - \underbrace{\nabla.(f\overline{\mathbf{u}})}_{planet.\ vort.\ adv} + \underbrace{\frac{J(P_b,h)}{\rho_0}}_{BPT} + \underbrace{\mathbf{k}.\nabla \times \frac{\tau_{wind}}{\rho_0}}_{wind\ curl} - \underbrace{\mathbf{k} \cdot \nabla \times \frac{\tau_{bot}}{\rho_0}}_{BDC} + \underbrace{D_\Sigma}_{horiz.\ diffusion} + \underbrace{A_\Sigma}_{NLA}
\tag{1}
$$

where the vorticity $\Omega$ is the curl of the vertically integrated components of the velocity between the bottom and the surface:

$\Omega = \mathbf{k} \cdot \nabla \times \overline{\mathbf{u}}$, with $\mathbf{u} = (u,v)$ the velocities in the $(x,y)$ direction. The overbar defines a vertically integrated quantity:

$$
\overline{u} = \int_{-h}^{\eta} u\,dz
\tag{2}
$$

with $\eta(x,y,t)$ the free surface height and $h(x,y)$ the topography. It is possible to decompose the planetary vorticity advection $-\nabla.(f\overline{u}) = -\beta V - f\frac{\partial \eta}{\partial t} \approx -\beta V$, with $V$ the vertically integrated meridional component of velocity, if we consider a mean over a long enough time period such that $\frac{\partial \eta}{\partial t} \approx 0$.

The non linear term can be written as:

$$
A_\Sigma = -\frac{\partial^2(\overline{vv} - \overline{uu})}{\partial x \partial y} - \frac{\partial^2 \overline{uv}}{\partial x \partial x} + \frac{\partial^2 \overline{uv}}{\partial y \partial y}
\tag{3}
$$

The expression for $A_\Sigma$ is similar to the one shown in Schoonover et al. (2016) (their equation (2)) but in our case, the integration between -h and $\eta$ allows their last term to cancel out with a residue from the inversion of the time derivative and the vertical integral in the rate term. The bottom pressure torque J($P_b$,h) is the Jacobian of the bottom pressure and the depth of the topography. It encompasses the effects of the varying topography on the flow, and is known to play a key role in the 200 barotropic vorticity balance of the SPG. In an idealized case of a geostrophic current flowing along a topography in free-slip condition, the BPT can be written $\frac{J(P_b,h)}{\rho_0} = f u_b \cdot \nabla h$ where $\rho_0$ is the mean reference density and the subscript $b$ denotes a field at the bottom. Given the kinematic condition at the bottom: $-u_b \cdot \nabla h = w_b$, the BPT can be written $\frac{J(P_b,h)}{\rho_0} = -f w_b$, which highlights the relation between the BPT and vortex stretching when the flow crosses an isobath.

The barotropic vorticity terms have already been computed for the North Atlantic using different models (OCCAM, ECCO, UVic ESCM, POP) at different resolutions ($1.8° \times 3.6°$, $1°$, $0.25°$, $0.2° \times 0.4°$, $0.1°$) in Hughes and De Cuevas (2001), Spence et al. (2012), Sonnewald et al. (2019), and Yeager (2015). Their major result is that the barotropic vorticity balance in the subtropical and subpolar gyres is at first order a balance between $\beta$V, $\nabla \times \frac{\tau_{wind}}{\rho_0}$, and $\frac{J(P_b,h)}{\rho_0}$.



In the subtropical gyre, the barotropic vorticity balance is close to a Sverdrup balance away from the boundaries ($\beta V \approx$

$\nabla \times \frac{\tau_{wind}}{\rho_0}$), while the closure of the northward branch of the gyre at the western boundary is done primarily through BPT ($\beta V \approx \frac{J(P_b,h)}{\rho_0}$) (Schoonover et al., 2016).

The barotropic vorticity balance in the SPG is slightly more complex due to the strong impact of the topography. Along the northern and western boundaries of the SPG, the first order balance is between meridional advection and BPT ($\beta V \approx \frac{J(P_b,h)}{\rho_0}$) (*e.g.* Hughes and De Cuevas (2001), their Figure 4; Yeager (2015), their Figure 1), with a significant impact of the wind only

in the northern part of the gyre along the Greenland coast. When the resolution of the model is increased from $1°$ to $0.1°$ in Yeager (2015), the main balances stay qualitatively similar, showing a modest effect of the eddies. Using a shallow water model with higher resolution ($1/20°$), Wang et al. (2017) illustrates the importance of the NL term in the dynamics of specific regions such as the Gulf Stream and the recirculation gyres. The viscous torque decreases in the boundary currents due to the lower viscosity of their model.

**4.2 Spatial scales of the vorticity balance**

In our simulations, the BPT balances the advection of vorticity at leading order everywhere in the domain (Fig. 6). This is qualitatively different from the vorticity balances shown in Yeager (2015), but it is similar to the results of Gula et al. (2015) in the Gulf Stream region with the same ocean model and a similar horizontal resolution. This highlights the fact that locally the flow is able to follow isobaths due to an equilibrium between the NL term (making the flow cross isobaths) and the bottom

pressure anomaly.

Both terms exhibit small scales related to topographic features, but with a high degree of cancellation between each other. The sum of the BPT and NL terms (Fig. 6 (c) is often an order of magnitude smaller than the amplitude of the terms considered individually and exhibits patterns and amplitudes matching the advection of planetary vorticity. This cancellation is also clear in Wang et al. (2017), their Figure 3, where the transport driven by mean flow advection balances the one driven by the BPT,

both having amplitudes larger than the wind stress curl-driven transport.

To facilitate the interpretation of maps of NL and BPT terms, the impact of small topographic scales has to be reduced by smoothing with a large enough length scale. NL terms in particular are expected to be smoothed out on scales larger than 1-2° (Hughes and De Cuevas, 2001). Figure 7 shows all terms smoothed with a gaussian kernel of $1°$ radius. Even with such smoothing, the BPT and NL terms are still significantly larger than the corresponding results from the $0.1°$ simulation of

Yeager (2015). However, their sum $\frac{J(P_b,h)}{\rho_0} + A_\Sigma$ (Fig. 7 (f)) is of the same order of magnitude than the $\beta$V (Fig. 7 (a) ) and the Bottom Drag Curl (BDC, Fig. 7 (e)).

The curl of the wind stress in POLGYR has the same pattern and amplitude than in Yeager (2015). It is mostly positive with the strongest signal on the Eastern coast of Greenland. The amplitude of the $\beta$V term is slightly stronger in our model than in coarser resolution simulations. In the simulations of Hughes and De Cuevas (2001) and Yeager (2015), the patterns of the

$\beta$V term seems to indicate much wider currents. Here, the patterns correspond to thinner and more intense currents, closely following the continental slopes, in agreement with the observations.




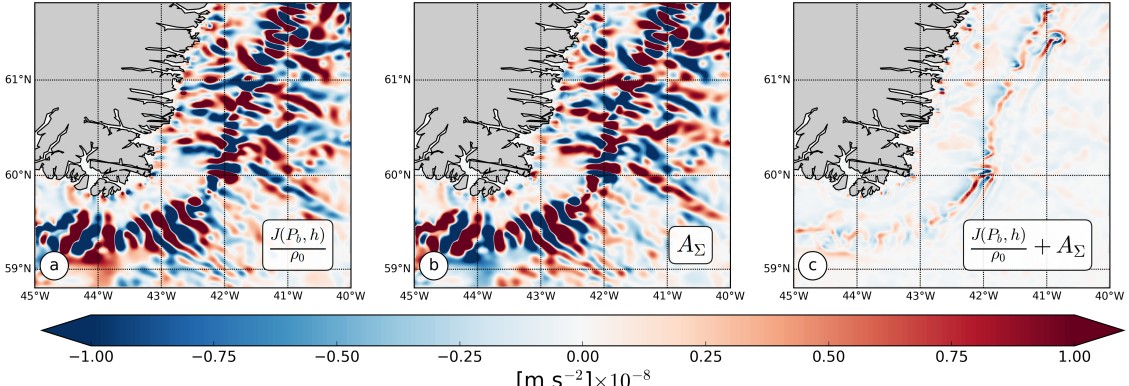

**Figure 6.** Time mean (a) bottom pressure torque, (b) non-linear terms, and (c) sum of the two for Eastern Greenland in the 2-km North Atlantic subpolar gyre simulation.

In our simulations, the amplitude of the viscous torque, due to the horizontal viscosity of the model ($D_\Sigma$), is very small, while the amplitude of the BDC is comparable to the $\beta V$. This is opposite to the results of the $0.1°$ POP simulation of Yeager (2015). In fact, their viscous term is qualitatively similar in pattern to the bottom drag curl in our simulation. The boundary conditions

near the topography are quite different in the two models due to the different vertical coordinates. The z-levels coordinates have vertical walls between each level, with parameterized lateral viscosity, which explains the pattern in Yeager (2015). The $\sigma$-levels coordinates have no lateral boundary conditions and friction on the topographic slopes is only parameterized as a bottom drag. The amplitude of the BDC is however stronger in our simulation than the viscous term in Yeager (2015) and seems to play a important role in balancing the BPT and $\beta V$ terms over the shelf and on the upper part of the continental slope

along the northern and western boundaries of the gyre.

### 4.3 Link between barotropic vorticity balance and bottom velocities

As explained previously, the bottom pressure torque J(P$_b$,h) can be identified with a bottom vortex stretching term: $\frac{J(P_b,h)}{\rho_0} = f u_{gb} . \nabla h = -f w_{gb}$, where $u_{gb}$ is the horizontal geostrophic bottom flow.

The computation of the BPT in Spence et al. (2012) is performed by directly estimating the term $-f w_b$, where $w_b$ is the

vertical velocity at the bottom. However this estimation does not take into account the presence of an ageostrophic component of the velocity at the bottom, in particular the Ekman component of the velocity due to the bottom drag. The same computation in our model leads to the results of Fig. 8(b), which are very different from the actual bottom pressure torque (Fig. 8 (a)). It gives results quite similar to Spence et al. (2012) with positive signals - implying downwelling of bottom currents - over most of the boundaries of the gyre. But this downwelling is a result of the Ekman currents oriented to the left of the main bottom

geostrophic currents, which are flowing with the shallower topography on their right around the gyre.





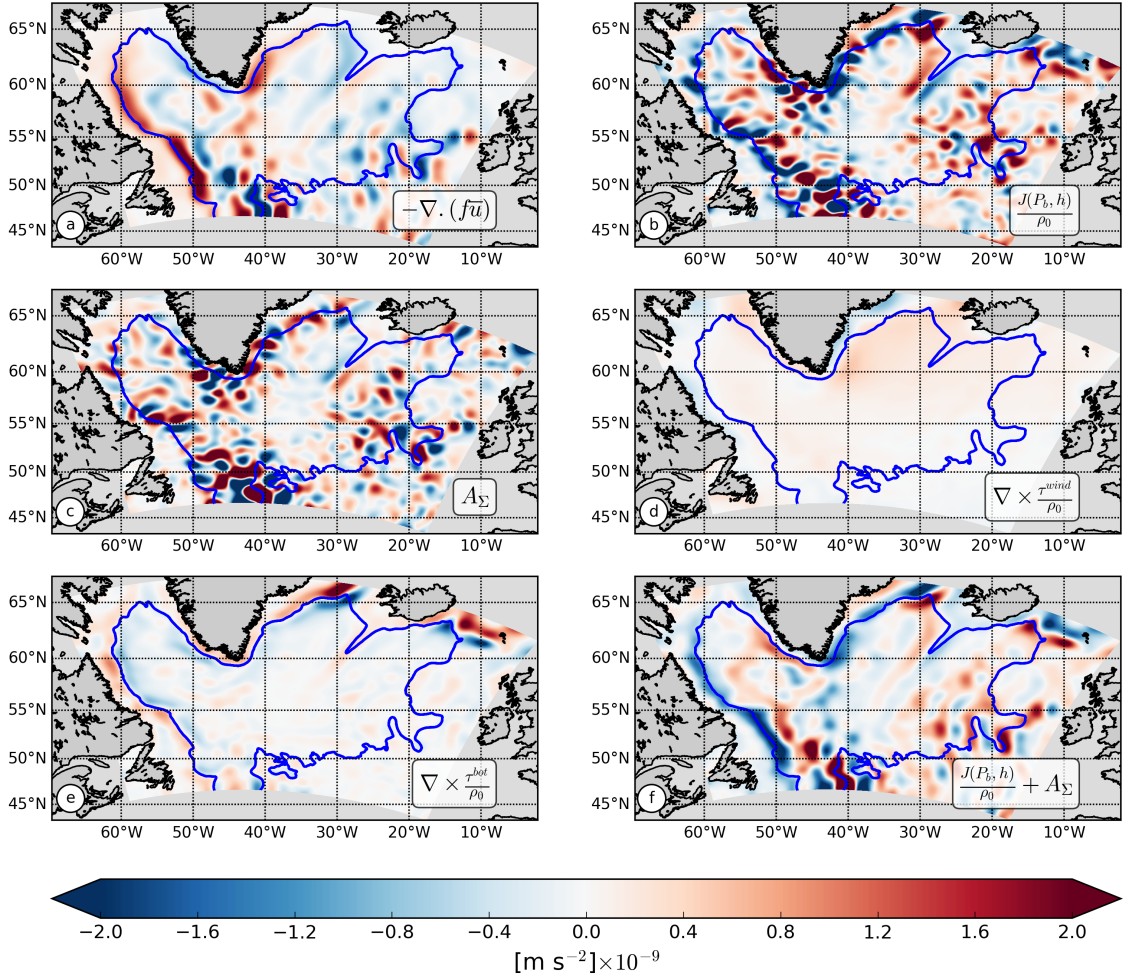

**Figure 7.** Time mean of the planetary vorticity (a), bottom pressure torque (b), non linear terms (c), wind stress curl (d), and bottom drag curl (e). As bottom pressure torque and non-linear terms are canceling each other their sum is plotted in (f). The fields have been smoothed using a kernel of $1°$ radius. The blue contour represents the limit of our shelf area and is the -3 Sv barotropic streamline

Following Mertz and Wright (1992) and Yeager (2015), the BPT can be further decomposed into:

$$\frac{J(P_b, h)}{\rho_0} = f u_{gb}.\nabla h = \frac{f}{h}\overline{u_g}.\nabla h + h(JEBAR) \tag{4}$$

which illustrates that the bottom geostrophic currents that appears in the expression of BPT are the sum of a vertically averaged part and a baroclinic part directly related to the JEBAR term. The term $\frac{f}{h}\overline{u_g} \cdot \nabla h$ highlights regions where the depth-averaged flow is crossing isobaths, and the $h(JEBAR)$ term where the baroclinic effects are playing a role to decouple the bottom flow from the barotropic flow through the geostrophic shear. In Fig. 8 (c) the geostrophic velocity has been computed from the thermal wind balance referenced at the surface.





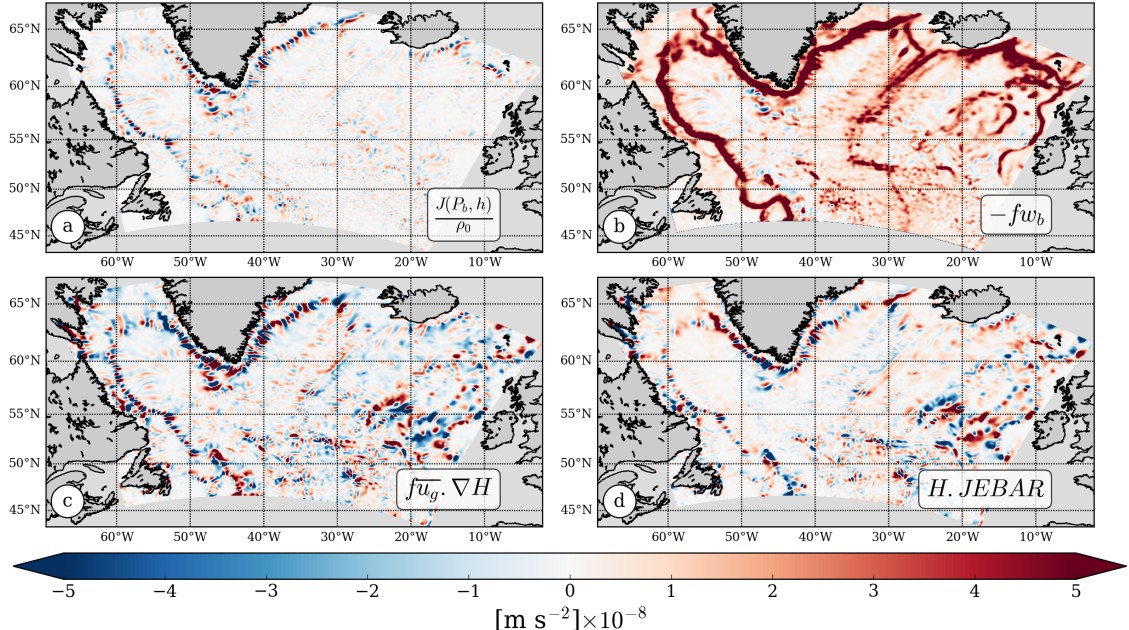

**Figure 8.** (a) Bottom Pressure Torque, (b) $-fw_b$, (c) $\frac{f}{h}\overline{u_g}\cdot\nabla H$, and (d) $H\cdot JEBAR$ for the 2-km North Atlantic subpolar gyre simulation smoothed with a 25 km Gaussian Kernel .

Along the continental slopes, on the western and northern part of the gyre, the flow is close to barotropic and the $\frac{f}{h}\overline{u_g}\cdot\nabla H$ term has similar patterns and amplitudes than the BPT. This contrasts with results from Yeager (2015), who found that the

$h(JEBAR)$ term was almost an order of magnitude larger than the BPT in these regions. However over the southern and eastern part of the gyre, it is clear that the structure of the flow is much more baroclinic and the $\frac{f}{h}\overline{u}\cdot\nabla h$ and $h(JEBAR)$ terms are both an order of magnitude larger than the BPT.

## 5   Integrated vorticity balance for the shelf, slope and interior of the gyre

### 5.1   Gyre integrated barotropic vorticity balances

The maps of the barotropic vorticity terms, with various degrees of smoothing, can help identify the locally dominant terms, but do not enable us to identify the important balances at the gyre scale. Spatial integrations are performed inside different gyre contours (Fig. 9) to better understand the main contributions to the circulation of the subpolar gyre.

We distinguish the shelf area from the gyre using a contour of barotropic streamfunction of -3 Sv. This contour is chosen because it corresponds to the largest possible closed contour of the barotropic streamfunction. We can check that the term

$-\nabla.(f\overline{u}) \approx -\beta V$ integrates to zero over such a contour (Fig. 9 (c)). The shelf thus defined corresponds to an area with a mean depth of 290 meter and is extending from the South of Iceland to Flemish cap (blue area in Fig. 9 (b)).





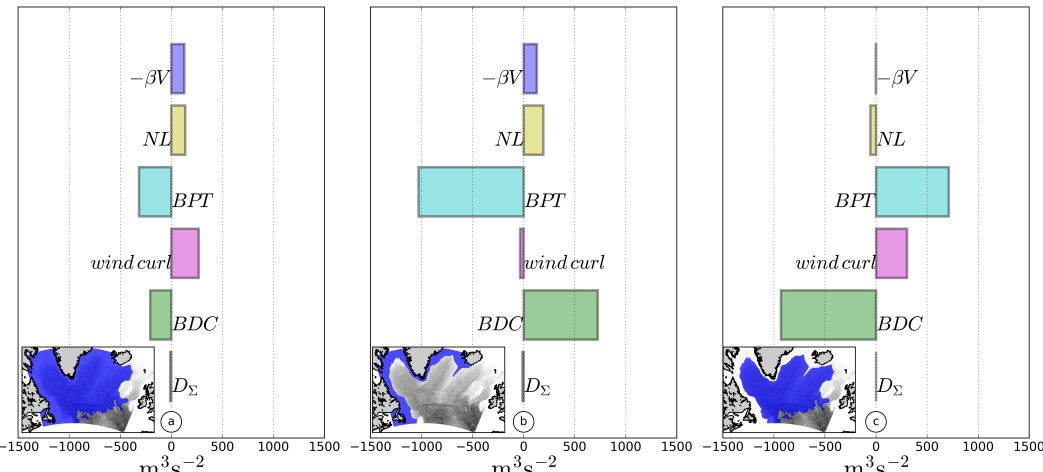

**Figure 9.** Integration of the barotropic vorticity terms over the SPG including or excluding the shelf area (respectively (a) and (c)). The Subpolar gyre area without the shelf corresponds to the -3 Sv contour. The shelf balance is plotted in (b).

When integrated inside the -3 Sv contour (which means excluding the shelf area, Fig. 9 (c)), the main sources for the cyclonic circulation of the gyre are the wind and the BPT. They are balanced by the BDC. The wind input does not contribute much locally (Fig. 7), but becomes significant when integrated spatially over the whole gyre. The BPT is the major source of positive

vorticity and helps the flow move cyclonically around the gyre. The BDC and NL terms act as sinks of vorticity, but the NL term is much smaller than the BDC. The BDC is very intense where the current flows close to a steep topography, as in the case of the Labrador Current (LC) and the West Greenland Current.

When integrated over the whole gyre (Fig. 9 (a)), the balance is slightly different. The wind is still a major contributor for the cyclonic circulation and the BDC still represents the major sink of vorticity. However, the NL term replaces the BPT as

a source of cyclonic vorticity for the gyre. In this interpretation, both the wind and the NL term forces the gyre cyclonically, while the BDC and BPT balance this input.

The difference between the two balances is highlighted by looking at the balance in the region in-between the two contours, which covers the upper slope and the shelf. It corresponds to a balance between BPT, NL and bottom drag. This balance is close to the one described in Csanady (1978) and evokes a buoyancy driven flow in this area (Chapman and Beardsley, 1989). Indeed,

with a switch to $(n,s)$ coordinates system with $n$ the right handed coordinates (here oriented toward shallower water) and $s$ the distance along flow, the BPT can be written $\frac{J(P_b,h)}{\rho_0} = -\frac{\partial P_b}{\partial s}\frac{\partial h}{\partial n}$. A negative value of BPT then means $\frac{\partial P_b}{\partial s} < 0$ corresponding to a buoyancy driven current.





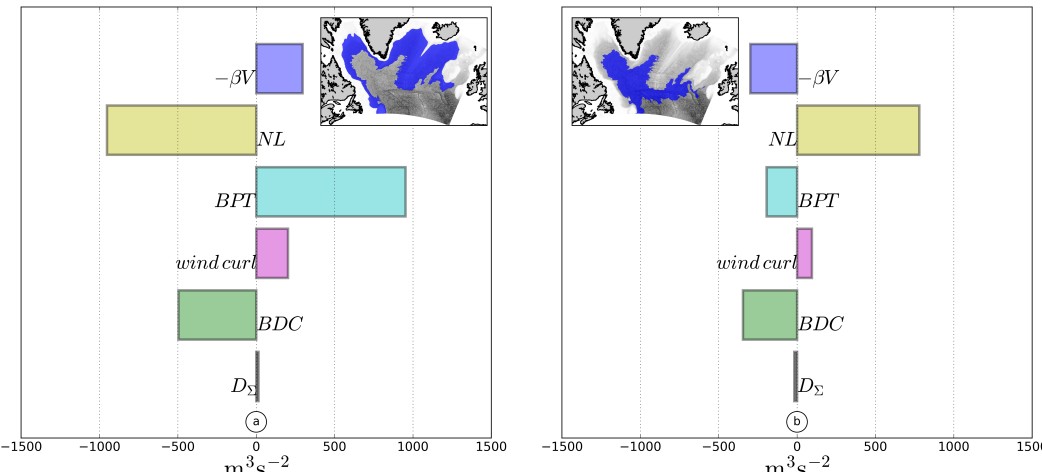

**Figure 10.** Integration of the barotropic vorticity terms in the slope area (a, defined between the barotropic streamfunction contour -3 Sv and the 3000-m isobath) and interior (b).

## 5.2 Barotropic vorticity balance in the interior of the gyre

It is clear from the patterns of the different terms of the barotropic vorticity balance that the local balances over the boundary currents are very different than what is happening in the interior of the gyre. The classical picture of a gyre interior (far from the boundaries) in a quasi-Sverdrup balance that applies in the subtropical gyre, does not seem to apply anywhere in the SPG.

To better understand what drives the interior part of the subpolar gyre, we further divide the domain into an interior and a boundary part, as represented in Figu. 10. The two domains are defined using the -3 Sv line as previously, and the 3000 m isobath. What is between the -3 Sv line and the 3000 m isobath is considered as the slope region and the rest is considered as the interior area. The choice of the 3000 m isobath is somehow subjective but the results are not sensitive to the choice of a specific isobath.

In the slope region, the main source of cyclonic vorticity is the BPT. The curl of the wind and the $\beta$V are also positive. The strongly negative NL term indicates advection of cyclonic vorticity outside of this domain toward the shelf or the gyre interior.

In the interior, the NL term represents the major contribution to the cyclonic circulation. It is balanced by the BDC, the BPT and the $\beta$V terms. Contributions from the BDC are of similar magnitude in the interior and the slope area. The wind input of vorticity is smaller than in the slope region, as the major wind source of vorticity is located along the Greenland area (Fig. 7 (d)) and not uniformly distributed over the gyre. It confirms that the gyre interior in not in Sverdrup balance at the first order, which would imply a dominant balance between a negative $\beta$V and a positive input from the curl of the wind stress, but is driven instead by nonlinear effects. The comparison between balances in the interior and slope regions indicates that the NL term helps to redistribute vorticity from the boundary toward the interior of the gyre.




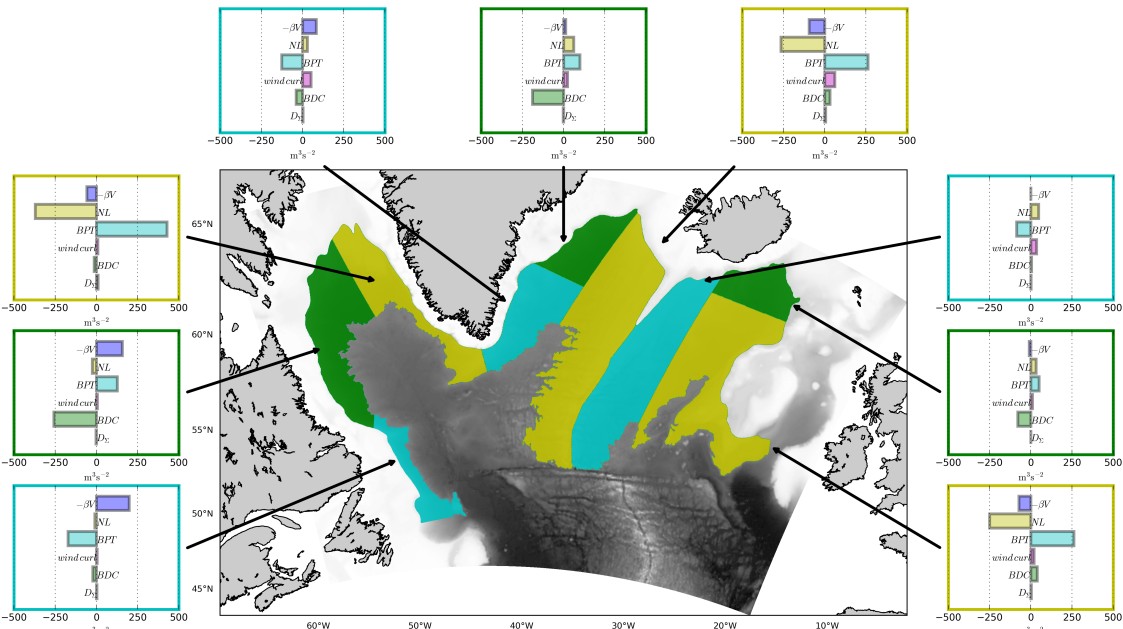

**Figure 11.** Barotropic vorticity balance integrated over different parts of the gyre along the slope

## 5.3 Balance in the slope area

The main source of cyclonic vorticity inside the gyre is related to the NL term, which helps transferring the vorticity from the boundary toward the inside. But which boundary regions are the main contributors of vorticity to the interior?

Several type of regions can be identified by looking at the dominant terms in the barotropic vorticity balance (Fig. 11):
The western boundary areas in cyan, which include the Western Labrador Sea (WLS), Eastern Greenland (EG) and Eastern Reykjanes Ridge (ERR); the eastern boundary regions in yellow, which include the Western Greenland (WG), the Western Reykjanes Ridge (ERR) and the eastern part of the Iceland Basin; and the Northwest regions in green, which include the extension of the Denmark Strait and Iceland Scotland overflows, and the northwestern part of the Labrador Sea.

The barotropic vorticity balance in the western boundary areas (cyan in Fig. 11) is close to the typical equilibrium of Western
Boundary Currents (WBC) (Schoonover et al., 2016; Gula et al., 2015) with an equilibrium between the planetary vorticity and the BPT. For the WLS, the deviation from WBC dynamics is small and is related to a bottom drag signal. We excluded the Southern part near Flemish Cap (48° N, 46° W) (not shown) where the dynamics is driven by a positive input of planetary vorticity and BPT balanced by the NL term. The case of the ERR is slightly different with no net meridional transport in this area. The main input of vorticity is provided by the NL term, which is related to inertial effects from the current following
the Iceland Shelf. In this area the input of positive vorticity is mainly balanced by topography and the drag corresponding to a local dissipation of vorticity. From this we can infer that western boundary areas do not provide cyclonic vorticity to the gyre interior.





Three regions (green in Fig. 11) have in common a dominant contribution from the bottom drag. Vertical sections of the mean along-stream current (Fig. 12 (a),(c),(e)) in these areas reveal strong intensified bottom current (especially near the Iceland

shelf and the Denmark Strait). In comparison, WBCs have a more surface intensified structure with reduced amplitudes near the bottom (Fig. 12 (b),(d),(f)). In Fig. 12, vorticity balances are indicated. They differs from Fig. 11 because the integration is restricted to the boundary current, excluding recirculations. In Fig. 12 (a),(c),(e) the BPT amplitudes are reduced (and even change sign) compared to Fig. 11. This reflects the sensitivity of the vorticity balance on the location of the boundary on the continental slope. The -3 Sv contour used in Fig. 11 does not coincide everywhere with the top of the continental slope used in

Fig. 12.

The dynamics in the extension of the Denmark Strait and Iceland Scotland overflows is a balance between the NL term and BDC, while in the Northwestern Labrador sea, the BDC balances the $\beta$-effect. As the BDC is the main sink of vorticity and only acts locally, no advection of positive vorticity toward the inside of the gyre can come from these locations.

In Eastern boundary regions (yellow in Figure 12), most of the cyclonic vorticity is provided by flow-topography interactions

through the BPT and is balanced by the NL term. These regions are located where a strong eddy activity is observed (Figure 5), which might be responsible for the high amplitude of the NL term. This negative NL signal implies an export of positive vorticity toward either the shelf or the gyre interior.

## 6   Characterisation of the nonlinear term

The NL term is locally important and balances the bottom pressure torque at small scales (Fig. 6). When integrated over the

gyre it plays a role in exporting cyclonic vorticity from the boundary toward the interior of the gyre. The NL term is however quite difficult to interpret as many processes are hidden inside the vertical and time integrals.

By decomposing the velocity in a barotropic and baroclinic part ($u = \overline{u} + u'$) the NL advection term can be written as:

$$A_\Sigma = \underbrace{A(\overline{u}, \overline{v})}_{A_\Sigma^{bt}} + \underbrace{A(u', v')}_{A_\Sigma^{bc}} \tag{5}$$

where the barotropic part can be written as $A(\overline{u}, \overline{v}) = \overline{u}\Omega_x + \overline{v}\Omega_y$ which is the advection of the barotropic vorticity by the

barotropic flow.

We show these terms integrated over the slope area and interior (same as Fig. 10) in Fig. 13. Over the slope area, both terms are negative and contribute to export cyclonic vorticity. The barotropic part is much larger than its baroclinic counterpart and export most of the vorticity, as can be expected from the barotropic structure of the currents over the slope. In the interior, both terms are positive, corresponding to an input of cyclonic vorticity for the interior (Fig. 13), but the NL term is evenly divided

between its barotropic and baroclinic contributions. The North West corner provides about half of this baroclinic NL input, while the remaining part comes mostly from the South-Eastern boundary. The exchange of barotropic vorticity is only due to the barotropic NL term between the slope region and the interior.





**Figure 12.** Vertical section of the mean along-stream current near Iceland shelf (a), Eastern Reykjanes Ridge (b), Denmark Strait (c), Eastern Greenland (d), Northern Labrador Current (e), and Southern Labrador Current (f). Red solid lines and green dashed lines are velocity and isopycnal contours, respectively, while the black dashed line is the limit of integration near the shelf. The black contour on the topography map represents the area on which barotropic vorticity terms are integrated





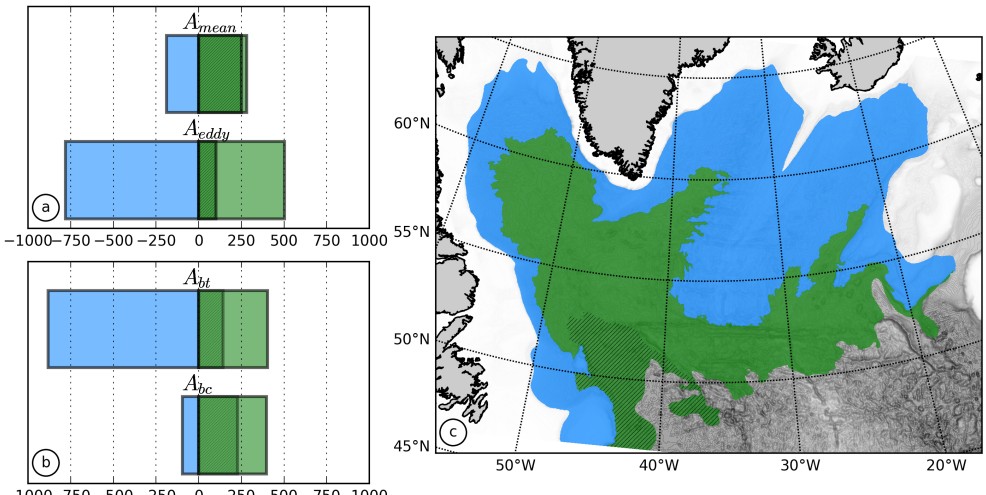

**Figure 13.** Integration of the Non linear term in the slope (c,blue) and interior area (c,green) for the mean-eddy decomposition (a)and the barotropic-baroclinic decomposition (b). The hacthes are the contribution from the North Western Corner .

It is also possible to decompose the NL term into a time mean and eddy part by writing $u = \langle u \rangle + u^*$ where $\langle \bullet \rangle$ is the time average and the star denotes the fluctuation part. By putting this in the non linear operator $A_\Sigma$ we have:

$$\Sigma(u,v) = \underbrace{A_\Sigma(\langle u \rangle, \langle v \rangle)}_{A_\Sigma^{mean}} + \underbrace{A_\Sigma(u^*, v^*)}_{A_\Sigma^{eddy}} + \underbrace{\langle 2\frac{\partial^2 \overline{\langle v \rangle v^*} - \overline{\langle u \rangle u^*}}{\partial xy} + \frac{\partial^2 \overline{\langle u \rangle v^*} + \overline{\langle v \rangle u^*}}{\partial xx} - \frac{\partial^2 \overline{\langle u \rangle v^*} + \overline{\langle v \rangle u^*}}{\partial yy} \rangle}_{\varepsilon} \tag{6}$$

The $\varepsilon$ part is the residue of the cross product and its value is negligible compared to both the mean and eddy parts.

When integrated over the slope area (Fig. 13), the eddy component dominates over the mean one. In the interior area, the supply of barotropic vorticity is also mainly due to the eddy component but the mean component contributes about a third of the total. Almost all of this mean signal is coming from the North West corner, consistent with Wang et al. (2017), while the eddy part is dominant over the rest of the interior.

We can identify several processes providing cyclonic barotropic vorticity to the subpolar gyre. The most important is the eddy contribution coming from the boundary area that is associated with a barotropic contribution. Barotropic vorticity is also provided through a mean-baroclinic signal coming from the NWC. In comparison, in the lower resolution simulation (not shown) most of the vorticity is advected inside the gyre by mean-barotropic processes but the amplitude of the NL term is cut by half.

## 7   Summary and Conclusions

We have studied the dynamics of the North Atlantic Subpolar gyre in a numerical model with, for the first time, terrain following coordinates and a mesoscale-resolving resolution ($\Delta x \approx 2$ km). The combination of the high resolution with $\sigma$-levels allows


us to better resolve the effects of the mesoscale turbulence and of the complex bottom topography. The representation of the

mean currents and their variability is improved compared to previous simulations with coarser resolution. In particular, the

simulations produce realistic levels of mesoscale turbulence at the surface and in the interior, as seen from comparisons of

eddy potential and kinetic energy with observations from Argo floats and surface drifters.

The role of the topography is essential in the SPG. This impact is reflected in the barotropic vorticity balance of the gyre

through the Bottom Pressure Torque. The Bottom Pressure Torque is sometimes interpreted as the effect of the vortex stretching

due to the bottom flow over topography, as expected for a predominantly geostrophic flow. However, we show here that the

ageostrophic effects, in particular due to the viscous bottom drag, are predominant at the bottom and the BPT cannot be

estimated from the bottom vertical velocity.

Barotropic vorticity balances are opposite in the shelf region compared to the inside of the gyre. The main balance in the

shelf region is between a negative bottom pressure torque and a positive bottom drag, which is typical of a buoyancy driven

current. Inside the gyre, the inputs of positive vorticity from the BPT and the wind curl, are balanced by the bottom drag curl.

The important role played by the bottom drag and the weak role played by the viscous torque, compared to other models, is

related to the choice of $\sigma$-level coordinates and high horizontal resolution.

The bottom pressure torque has a large amplitude where boundary currents flow along the steep continental slope. It is the

main term balancing the meridional transport of water in western boundary currents, except for some regions with dense water

overflows where the bottom drag curl can become predominant. On the eastern (northward flowing) boundary currents, the

strong input of positive vorticity by the bottom pressure torque is balanced by the non linear term. The nonlinearities, which

are essentially due to the eddying activity, allow advection of the positive vorticity from the boundary toward the interior of

the gyre. The North Western Corner is also instrumental in feeding positive vorticity to the gyre interior through its southern

boundary, mostly through time-mean baroclinic fluxes.

The nonlinear term is the main forcing for the interior part of the gyre, overcoming the effects of the wind curl and bottom

pressure torque. This is putting forward the failure of the classical Sverdrup balance or even of a topographic Sverdrup balance

in the interior of the Subpolar gyre, and emphasizing the importance of the inertial effects to obtain a more realistic Subpolar

gyre circulation.

*Data availability.* The datasets ANDRO and NOAA are respectively available online at https://www.umr-lops.fr/Donnees/ANDRO and

https://www.aoml.noaa.gov/phod/gdp/interpolated/data/subset.php.

*Author contributions.* MLC designed the setup and carried out the experiment. MLC and JG analysed the output of the simulation. All

authors participated in the writing and editing of the article.





*Competing interests.* The authors declare that they have no conflict of interest.

*Acknowledgements.* M Le Corre is supported by UBO and Région Bretagne through ISblue, Interdisciplinary graduate school for the blue
planet (ANR-17-EURE-0015) and co-funded by a grant from the French government under the program "Investissements d'Avenir". JG
is supported by UBO and AM Treguier by CNRS. Simulations were performed using HPC resources from GENCI-TGCC (grant 2018-
A0050107638) and from DATARMOR of "Pôle de Calcul Intensif pour la Mer" at Ifremer, Brest, France. Model outputs are available upon
request.





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
