# Peer review of "Barotropic vorticity balance of the North Atlantic subpolar gyre in an eddy-resolving model"

_Ocean Science, 2019_

## Referee Comment (RC1) · Anonymous Referee #1 · 17 Nov 2019

Comments on "Barotropic vorticity balance of the North Atlantic subpolar gyre in an eddy-resolving model" by Le Corre et al..

I have enjoyed reading this manuscript and consider it to be an excellent piece of work that I recommend for publication after minor revision. The important role played by the nonlinear terms in the dynamics of the subpolar North Atlantic was anticipated by Wang et al. (2017) but is comprehensively demonstrated in this manuscript using a very thorough and detailed analysis of the barotropic vorticity balance in a model for the subpolar gyre with kilometer resolution. The manuscript is an impressive piece of work and forces a change in our view on how the subpolar gyre in the North Atlantic is driven and maintained. I do, nevertheless, have one major comment for the authors to address in addition to a number of minor comments.

Major Comment

Abstract and elsewhere: The authors claim that the Northwest Corner is a source of vorticity through the non-linear terms for driving the subpolar gyre in the North Atlantic. But I do not see how this is possible dynamically. The problem is that information propagates westwards along potential vorticity contours – either lines of latitude, as in the formulation of the barotropic vorticity balance given by equation (1), or along f/H contours as in the formulation used by Wang et al. (2017) (see comment 10. below). In either case, it is not possible for a vorticity forcing applied at the northwest corner to influence the gyre interior. It seems to me, therefore, that it is the eastern boundary regions that are important for driving the gyre and not the Northwest corner. Unfortunately, one cannot appeal to non-linear advection to get around this problem. To be influential, the non-linear term must be important in the eastern part of the gyre or in the gyre interior itself.

Minor Comments:

1. Line 31: Why mention Munk (1950) but not Stommel (1948)? I would refer to both.
2. Line 35: An important role for the bottom pressure torque is also anticipated in the early, diagnostic model of Greatbatch et al. (1991) – their Figure 6.
3. Lines 51-52: Wang et al. also showed the importance of the nonlinear terms in the subpolar gyre for driving the so-called Lavender recirculation – see their Figure 2c.
4. Line 80: From Chelton et al., I would say that the radius of deformation for the 1$^{st}$ baroclinic mode has trouble exceeding 10 kms and certainly does not reach 20 kms – see their Figure 6.
5. Line 97: From Figure 2, the vertical grid does not look to be particularly bottom intensified?
6. Figure 3: It is not easy to see all the details in this figure – although I do not have specific suggestions for improvement.
7. Line 156-157: As noted above, Wang et al. find an important role for what they call "mean flow advection" for driving the Lavender recirculation along the slope around the Labrador Sea.
8. Paragraph beginning on line 165: Could refer to Brandt et al. (2004, JGR).
9. Line 175: How is EAPE defined? This should be given somewhere.
10. Line 186 and equation (1): Should mention that this is the vorticity equation for the vertically integrated flow. There is also an equivalent vorticity equation for the vertically averaged flow.

11. Lines 205-206: Do the acronyms for these different models get defined somewhere?
12. Line 218: …also the subpolar North Atlantic, as noted above (point 3).
13. Line 221: I would not say the "advection of vorticity" when you are referring to the nonlinear term. It is easy to confuse with the "advection of planetary vorticity".
14. Line 280: Should "over" be replaced by "within"? Actually, the integral of this term should be very close to zero by construction.
15. Line 294: My only objection here is that the Csanady paper uses dynamics linearized about a state of rest which means that the NL term plays no role, as could, perhaps, be made clearer. However, the comparison with the arrested topographic wave is certainly illuminating.
16. Figure 12: The dashed lines show isopycnal surfaces but which density is this? From the labelling, it must be a potential density of some kind. Please make clear.

Typos and language issues:

1. Line 18: "this" -> "these"
2. Line 19: "drives" -> "drive"
3. Line 31: "low" -> "weak"
4. Line 62: "coordinates" -> "coordinate" in both occurrences.
5. Line 65: "confronted" -> "compared"
6. Line 67: "analuze" -> "analyse"; "it more details" -> "it in more detail"
7. Caption to Figure 2: "of the Irminger Sea" -> "in the Irminger Sea"
8. Line 147: "coherent" -> "consistent"
9. Line 148: "which" -> "who"
10. Line 159: "center" -> "centre of the"
11. Line 160: "North" -> "northward"
12. Line 233: "Gaussian" -> "Gaussian"
13. Line 237: "than" -> "as"
14. Line 303: "Figu." -> "Fig."
15. Line 311: "along the Greenland area" -> "near Greenland"
16. Line 317: "transferring" -> "transfer"
17. Line 336: "differs" – "differ"
18. Caption to Figure 13: "hacthes" -> "hatches"
19. Line 388: "inside" -> "interior"
20. References: The reference to Le Bras et al. has some repeat. Please check all the references!

---

## Referee Comment (RC2) · Anonymous Referee #2 · 19 Dec 2019

This paper is a thorough and well-written breakdown of the barotropic vorticity balance of the subpolar gyre. I enjoyed reading the paper and found its arguments to be convincing. The model itself is impressive and Section 3 is a convincing validation of its circulation. I have a few comments that could be addressed by the authors, but otherwise find the paper impressive and worth publishing.

My most significant comments are numbered below. More minor comments follow.

1) Like the rest of the paper, the introduction is thorough and well-written. I think it it could do a better job of emphasising the novelty of the paper a little more strongly, particularly with regard to the vertical coordinate/nested domain and thorough analysis/breakdown of the barotropic vorticity equation.

[Figure]

2) The model run is relatively short, although page 3 does discuss the spinup. Whilst this is probably sufficient to equilibrate the barotropic mode, there is a link to the baroclinic mode via the JEBAR term. Is the baroclinic mode properly spunup? If it isn't, the authors should discuss any impact this might have on their argument.

Minor comments, typos, etc

line 3 : "this dynamics has" -> "these dynamics have"

lines 9-10 : "the topographic Sverdrup balance cannot describe the dynamics in the interior". which it probably shouldn't. I'd expect the flat bottomed Sverdrup balance to dominate here.

line 18 : "this complex" -> "these complex"

line 31 : It would also be appropriate to mention bottom friction and Stommel (1948) here, as bottom friction is discussed later, e.g. Fig. 7e.

Section 2 : No mention of horizontal viscosity or diffusivity, although the rest of the section is very thorough.

line 132 : rogue "?" in brackets.

line 235 & 243 : There's really only a few locations where the BDC is large. It seems largely the case that betaV balances the residual of the NL and BPT terms.

line 292-297 : This goes past a little too quickly for me. Without further reading, or a more in depth description, I find it difficult to make the link between the gradient of the bottom pressure and the nature of the flow's driving force.

line 303 : "Figu. 10"

line 317 : "transferring" -> "transfer"

Figure 13 caption : "hacthes"?

[Figure]

---

## Author Comment (AC1) · 25 Feb 2020

Dear referee,

Please find in attached the reponses to your comments and suggestions, and a new version of the manuscript

Kind regards,

Mathieu Le Corre

Please also note the supplement to this comment:
https://www.ocean-sci-discuss.net/os-2019-114/os-2019-114-AC1-supplement.zip

---

## Author Comment (AC2) · 25 Feb 2020

Dear referee,

Please find in attached the reponses to your comments and suggestions, and a newversion of the manuscript.

Kind regards,

Mathieu Le Corre

Please also note the supplement to this comment:
https://www.ocean-sci-discuss.net/os-2019-114/os-2019-114-AC2-supplement.zip

---

## Author Response (AR2)

[revised manuscript text omitted]

**Response to Referee #1**

We would like to thank the Referee for his/her constructive comments. We have taken into account all the points that were raised and we document the changes below.

Major comment :

*Abstract and elsewhere: The authors claim that the Northwest Corner is a source of vorticity through the non-linear terms for driving the subpolar gyre in the North Atlantic. But I do not see how this is possible dynamically. The problem is that information propagates westwards along potential vorticity contours – either lines of latitude, as in the formulation of the barotropic vorticity balance given by equation (1), or along f/H contours as in the formulation used by Wang et al. (2017) (see comment 10. below). In either case, it is not possible for a vorticity forcing applied at the northwest corner to influence the gyre interior. It seems to me, therefore, that it is the eastern boundary regions that are important for driving the gyre and not the Northwest corner. Unfortunately, one cannot appeal to non-linear advection to get around this problem. To be influential, the non-linear term must be important in the eastern part of the gyre or in the gyre interior itself.*

We thank the reviewer for this comment. The confusion comes from our definition of the gyre interior with the 3000-m isobath and the -3-Sv barotropic streamfunction contour (along the South Eastern edge).  Due to baroclinicity, this region includes the Northwestern Corner (NWC) which can  also be viewed as part of the subtropical gyre.
It is true there is no vorticity flux from the NWC to the subpolar gyre interior (excepted maybe by a small eddy component). However the vorticity balance of the region we have defined as subpolar (based on the barotropic streamfunction) is influenced by the NWC.
This is now made clear in  (l.387-390) :
« Barotropic vorticity is also provided through a mean-baroclinic signal located in the NWC. Our definition of the subpolar gyre, based on a barotropic streamfunction contour, includes a part of the NWC which is a complex transition region between the subtropical and the subpolar gyre. »

*1. Line 31: Why mention Munk (1950) but not Stommel (1948)? I would refer to both.*

A reference to Stommel (1948)  was added  on l.31

*2. Line 35: An important role for the bottom pressure torque is also anticipated in the early, diagnostic model of Greatbatch et al. (1991) – their Figure 6.*

A reference to Greatbatch (1991) was added on l.35

*3. Lines 51-52: Wang et al. also showed the importance of the nonlinear terms in the subpolar gyre for driving the so-called Lavender recirculation – see their Figure 2c.*

The case of the Lavender recirculation was added to the list of locations where the NL term is important (l.54)

*4. Line 80: From Chelton et al., I would say that the radius of deformation for the 1 st baroclinic mode has trouble exceeding 10 kms and certainly does not reach 20 kms – see their Figure 6.*

On the Southern edge of the domain the first radius of deformation is close to 20-km. To nuance our words we are now saying « first Rossby deformation radius remains below 10-km over most of the region »  (l.80)

*5. Line 97: From Figure 2, the vertical grid does not look to be particularly bottom intensified?* We replaced the section in the Irminger basin by one in the Labrador Sea where we think it is clearer.  We also added the variation of the grid spacing with depth along the vertical black line in (b).

[Figure]

*6. Figure 3: It is not easy to see all the details in this figure – although I do not have specific suggestions for improvement.*

We tried to improve the figure by changing the colormap to make the arrows more visible (p.6)

*7. Line 156-157: As noted above, Wang et al. find an important role for what they call "mean flow advection" for driving the Lavender recirculation along the slope around the Labrador Sea.*

Thank you for pointing this oversight. The following was added : « More recently Wang (2017) showed the importance of the mean flow advection in these circulations. ». (l.161)

*8. Paragraph beginning on line 165: Could refer to Brandt et al. (2004, JGR).*

Thank you for suggesting Brandt et al. (2004), the reference was added. (l.169)

*9. Line 175: How is EAPE defined? This should be given somewhere.*

The definition of EAPE is now added in the new equation (1) :
$$\text{EAPE} = \frac{-g}{2\rho_0}\langle z'\rho'\rangle$$
Where z' is the vertical isopycnal displacement, $\rho'$ the density anomaly associated with this displacement and $\langle . \rangle$ is the time average.
Also precisions on the EAPE version of Roullet et al. (2014) were added. (l.180-186)

*10. Line 186 and equation (1): Should mention that this is the vorticity equation for the vertically integrated flow. There is also an equivalent vorticity equation for the vertically averaged flow.*

We are now mentionning the two different versions for the barotropic equations and commenting their differences in the text. (l.195-199)

*11. Lines 205-206: Do the acronyms for these different models get defined somewhere?*

Acronyms are now defined along with references to previous studies using these models. (l.216-218)

*12. Line 218: ...also the subpolar North Atlantic, as noted above (point 3).*

Same as in point 3 (l.231-232)

*13. Line 221: I would not say the "advection of vorticity" when you are referring to the nonlinear term. It is easy to confuse with the "advection of planetary vorticity".*

In order to avoid confusion « advection of vorticity » was changed by « nonlinear term » (l.233)

*14. Line 280: Should "over" be replaced by "within"? Actually, the integral of this term should be very close to zero by construction.*

« Over » was replaced by « within ». Because of model discretisation the integral is not exactly zero but very close. (l.293)

*15. Line 294: My only objection here is that the Csanady paper uses dynamics linearized about a state of rest which means that the NL term plays no role, as could, perhaps, be made clearer. However, the comparison with the arrested topographic wave is certainly illuminating.*

A reference to Csanady (1997) about JEBAR effect on the shelf has been added. The NL term is only important along the Greenland shelf and is related to eddy-barotropic component suggesting eddy interaction between the shelf and the open ocean. On the Canadian shelf the NL term is small and is barely contributing to the dynamics, thus the use of linearized dynamics seems valid there. The part with the coordinate changes has been removed for clarity. (l.305-310)

*16. Figure 12: The dashed lines show isopycnal surfaces but which density is this? From the labelling, it must be a potential density of some kind. Please make clear.*

Indeed, we are talking about potential density referenced at the surface. This precision was added in the caption. (p.19)

Typos and language issues :

Typos and language issues were corrected.

**Response to Referee #2**

We would like to thank the Referee for his/her constructive comments. We have taken into account all the points that were raised and we document the changes below.

**Major comment :**

*1) Like the rest of the paper, the introduction is thorough and well-written. I think it could do a better job of emphasising the novelty of the paper a little more strongly, particularly with regard to the vertical coordinate/nested domain and thorough analysis/breakdown of the barotropic vorticity equation.*

We have slightly modified the abstract (l.5-6) and introduction (l. 63-65) to further emphasize the main novelty of the study.

*2) The model run is relatively short, although page 3 does discuss the spinup. Whilst this is probably sufficient to equilibrate the barotropic mode, there is a link to the baroclinic mode via the JEBAR term. Is the baroclinic mode properly spunup? If it isn't, the authors should discuss any impact this might have on their argument.*

[Figure]

To evaluate the equilibration of the model, we show the time series of the barotropic (top) and baroclinic energies (bottom) for NATL (green) and POLGYR (blue). The area of integration is the same and corresponds to the entire POLGYR domain. We can see that the amount of energy in the barotropic mode reaches a statistical equilibrium pretty fast, over about a year.

By 2001, the baroclinic mode is also close to equilibration in NATL. One additional year of spin up for the POLGYR nest from 2001 to 2002 allows the dynamics to adjust to the increase in resolution. The text has been modified to refer to both barotropic and baroclinic energies (l.94)

Minor comments :

Typos issues as in line 3- 18- 132- 303- 317- Fig13 were corrected

*lines 9-10 : "the topographic Sverdrup balance cannot describe the dynamics in the interior". which it probably shouldn't. I'd expect the flat bottomed Sverdrup balance to dominate here.*

We changed « topographic Sverdrup balance » to «Sverdrup balance » in the abstract (l.10)

*line 31 : It would also be appropriate to mention bottom friction and Stommel (1948) here, as bottom friction is discussed later, e.g. Fig. 7e.*

In addition to viscous effects, bottom friction was added along with a reference to Stommel (1948) (l.30-31)

*Section 2 : No mention of horizontal viscosity or diffusivity, although the rest of the section is very thorough.*

We have added the following in the model description : « We use no explicit horizontal viscosity or diffusivity and rely on third-order upwind-biased advection schemes, which include  an implicit hyperdiffusivity at the grid scale. » (l.103-105)

*line 235 & 243 : There's really only a few locations where the BDC is large. It seems largely the case that betaV balances the residual of the NL and BPT terms.*

l.235 :  We have added a sentence to make this more explicit in the text (l .248 in the revised version of the manuscript) : « More precisely, the  $\beta V$  term balances the sum  $\frac{J(P_b,h)}{\rho_0}+A_\Sigma$  over most of the domain, while the BDC locally plays a role in the shallow areas. »

l.243 : The sentence has been modified to avoid any misunderstanding  (l.255)

*line 292-297 : This goes past a little too quickly for me. Without further reading, or a more in depth description, I find it difficult to make the link between the gradient of the bottom pressure and the nature of the flow's driving force.*

We have modified this part and removed the explanation based on the coordinate change to make the interpretation clearer. (l.305-310)